# Regulation of Stanniocalcin Secretion by Calcium and PTHrP in Gilthead Seabream (*Sparus aurata*)

**DOI:** 10.3390/biology11060863

**Published:** 2022-06-04

**Authors:** Ignacio Ruiz-Jarabo, Silvia F. Gregório, Juan Fuentes

**Affiliations:** 1Centre of Marine Sciences (CCMAR), Campus de Gambelas, Universidade do Algarve, 8005-139 Faro, Portugal; sfgregorio@ualg.pt; 2Department of Physiology, Faculty of Biological Sciences, University Complutense Madrid, 28040 Madrid, Spain

**Keywords:** calcium-sensing receptor, corpuscles of Stannius, parathyroid hormone-related protein, stanniocalcin, *Sparus aurata*

## Abstract

**Simple Summary:**

Calcium regulation in body fluids is a fundamental process in vertebrates, which is exerted by a plethora of hormones. Stanniocalcin (STC) is a hypocalcemic hormone ubiquitously expressed in tetrapods; in bony fishes, it is produced mainly by specific glands called the corpuscles of Stannius. The present study described an ELISA method for the analysis of fish STC. Moreover, it also develops a methodology for ex vivo cultures of Stannius corpuscles of gilthead seabream (*Sparus aurata*). The results show a direct control of the production of STC by one calcium and two PTHrP (parathyroid-related protein, a hypercalcemic hormone) receptors. This study highlights the tight control of circulating calcium in vertebrates and shows the complexity of the processes involved.

**Abstract:**

Calcium balance is of paramount importance for vertebrates. In fish, the endocrine modulators of calcium homeostasis include the stanniocalcin (STC), and some members of the parathyroid hormone (PTH) family, such as the PTH-related protein (PTHrP), acting as antagonists. STC is ubiquitously expressed in higher vertebrates. In turn, bony fish exhibit specific STC-producing glands named the corpuscles of Stannius (CS). Previous studies pointed to a calcium-sensing receptor (CaSR) involvement in the secretion of STC, but little is known of the involvement of other putative regulators. The CS provides a unique model to deepen the study of STC secretion. We developed an ex vivo assay to culture CS of fish and a competitive ELISA method to measure STC concentrations. As expected, STC released from the CS responds to CaSR stimulation by calcium, calcimimetics, and calcilytic drugs. Moreover, we uncover the presence (by PCR) of two PTHrP receptors in the CS, e.g., PTH1R and PTH3R. Thus, ex vivo incubations revealed a dose-response inhibition of STC secretion in response to PTHrP at basal Ca^2+^ concentrations. This inhibition is achieved through specific and reversible second messenger pathways (transmembrane adenylyl cyclases and phospholipase C), as the use of specific inhibitors highlights. Together, these results provide evidence for endocrine modulation between two antagonist hormones, STC and PTHrP.

## 1. Introduction

Calcium is of paramount importance for many physiological processes in vertebrates, such as bone and scale mineralization, enzyme control, and regulation of plasma membrane permeability and excitability [1], among other functions. The delicate calcium balance is not subject to purely diffusive processes but is under strict endocrine regulation. In this sense, teleost fish primarily regulate calcium availability by the action of two antagonistic hormones, the parathyroid hormone-related protein (PTHrP) and the stanniocalcin (STC) [2,3,4]. The first is a hypercalcemic hormone [5], while the second has anti-hypercalcemic functions in fish [6].

Specific glands named corpuscles of Stannius (CS) located on the ventro-caudal side of the kidney secrete stanniocalcin in bony fish [7]. In mammals, stanniocalcin is expressed ubiquitously [8,9]. Besides being a hypocalcemic hormone, it is related to intestinal [10] and renal [11] phosphate reabsorption in fish and mammals, metastasis-related processes [12] through autocrine/paracrine routes [13], and transport of other ions rather than Ca^2+^ in zebrafish [14]. Increased calcium levels evoke the production and secretion of STC [15,16] mediated by a calcium-sensing receptor or CaSR [17,18]. However, more recent studies show that the CaSR modulates PTHrP, in addition to STC [19,20], as was described previously for the mammalian parathyroid hormone and calcitonin [21]. Moreover, STC and PTHrP hormones revealed antagonistic functions in fish regarding intestinal bicarbonate secretion [3] and carbonate precipitate formation in the fish intestine [22].

There are vital pieces of evidence concerning the regulation of both STC and PTHrP through a CaSR, a unique novelty receptor firstly described in bovine parathyroid [23]. Over the last two decades, the CaSR has received particular attention, especially in marine fish, due to its importance in maintaining the water and ionic balance [24]. As it belongs to the G protein-coupled receptor (GPCR) superfamily, it induces the activation of different types of G protein [25] when stimulated by calcium or other agonists, such as calcimimetics, or inhibited by calcilytics [26]. However, the second messenger routes activated through this membrane sensor are enormously variable, thus stimulating enzymes such as phospholipases (either PLC, PLA2, or PLD) and MAPK signaling cascades and inhibiting cAMP accumulation due to the inhibition of transmembrane adenylyl cyclases, tmAC [27].

Furthermore, membrane receptors coupled to G proteins mediate the actions of PTHrP [2,28]. Teleost fish have three PTHrP characterized receptors [29,30,31], with only two in our model species, the gilthead seabream (*Sparus aurata*) [32]. PTHrP receptor ligands can stabilize different conformations of the same receptor, thus leading to a differential activation pattern called “biased receptor signaling” [33], which means that one ligand can act differentially on multiple signaling pathways. In this sense, studies in humans, rats, and zebrafish single PTH/PTHrP receptors established that their coupling to the adenylyl cyclase (AC)-protein kinase A (PKA) signaling pathway and to the phospholipase C (PLC)-protein kinase C (PKC)-intracellular Ca^2+^ signaling pathway, may occur at the same time [2,28,31]. However, previous evidence indicates that the receptor PTH3R in gilthead seabream only activates the cAMP pathway [32] related to tmAC. This evidence indicates that the affinities to those second messenger routes could rely on the specific PTHrP receptor.

Provided that STC and PTHrP have antagonistic functions in most biological tests performed thus far, we aimed to show a functional interrelationship between STC and PTHrP. Further, we aimed to establish if PTHrP has a regulatory role on STC synthesis and secretion at the level of the corpuscles of Stannius and whether there is an intermediary role for the CaSR.

## 2. Materials and Methods

### 2.1. Animal Maintenance

Seabream (*Sparus aurata*) immature juveniles (84.2 ± 1.9 g body weight, mean ± SEM, *n* = 134) were obtained from commercial sources in the Algarve (Portugal) and transported to the Campus facility (CCMAR, Faro-Portugal). Before the experiments, fish were maintained for at least ten days in 500 L tanks in a recirculated seawater circuit (36–37‰) with a biological filter. The water temperature was maintained at 23–25 °C. The fish were exposed to a natural photoperiod (September-December), grouped in tanks at a density of 4.5 kg m^−3,^ and fed twice daily 1% body weight (wet fish weight, Sorgal S.A., São João de Ovar, Portugal; Balance 3). Animals fasted 24 h before the experiments, and there was no mortality during these processes. The experiments conducted comply with the guidelines of the European Union Council (86/609/EU). All animal protocols were performed under a ‘‘Group-1′’ license from the Direcção-Geral de Veterinária, Ministério da Agricultura, do Desenvolvimento Rural e das Pescas, Portugal.

### 2.2. Sampling

Fish were captured and anesthetized in 2-phenoxyethanol (0.1% *v*/*v*; P-1126, Sigma-Aldrich, St Louis, MO, USA). Blood samples were collected from the caudal peduncle with heparinized syringes, and plasma was obtained by centrifugation (3500× *g*, 5 min, 4 °C) and stored at −20 °C until analysis. Fish were sacrificed by decapitation, and the CS collected (their location in the caudal end of the kidney can be seen in Appendix A) for ex vivo culture (see below).

### 2.3. Development of an ELISA Method

Stanniocalcin protein levels were quantified by an in-house developed indirect antigen competitive ELISA method, similarly to what was previously described [34]. The assay is based on competition between free STC in standards, or samples and STC immobilized on microtiter plates for the STC-antibodies. Purified seabream STC [3] was employed to generate a specific rabbit antiserum [22], hereafter named Anti-STC. A range of assay optimization procedures was carried out, including assessment of assay buffers; the coating concentration of STC (from CS-homogenates, 0.5–10.0 μg prot well^−1^); titration of primary (1:1500–1:8000), and secondary antisera (1:2000–1:6000) to give a maximal optical density (OD490 nm) equal or higher than one and background not higher than 0.1 (OD490 nm); assay incubation times and washing conditions. ELISA validation included verification of parallelism between samples and the standard curve, the recovery rate of standards added to the assay, and the intra-assay and inter-assay variation coefficients.

Using the optimized procedure, ELISA was carried out in duplicate for standard and samples in 96-well plates (Ref. 655061, Greiner bio-one, Kremsmünster, Austria). In brief, wells were coated for 2 h at 37 °C with 2 μg prot 100 μL^−1^ well^−1^ of seabream CS-homogenates (in 0.6% *w*/*v* NaCl) diluted in bicarbonate buffer (50 mM NaHCO_3_/ Na_2_CO_3_ and 150 mM NaCl to give a final pH of 9.6). All the wells, including blank wells (without STC coating), were then blocked with 250 μL of 2% (*w*/*v*) bovine serum albumin (BSA) in PBS. Anti-STC (1:8000 dilution) was added to the plate along with either standard protein (range from 2000 to 52 ng mL^−1^) or samples (100 μL well^−1^) and incubated 10 min at room temperature with constant stirring, followed by incubation for 30 min at 37 °C. Anti-rabbit IgG peroxidase conjugate (100 μL well^−1^, 1:5000 dilution, A9169, Sigma-Aldrich) was added and followed by an incubation similar to the primary antibody. Excess reagents were removed between incubation steps by washing 4 × 5 min with PBS/Tween-20 (0.05% *v*/*v*). The color was developed with 200 μL OPD (o-phenylenediamine dihydrochloride, P9187, Sigma-Aldrich) for 13 min in the dark at room temperature. The reaction stopped by adding 50 μL well^−1^ 2 M H_2_SO_4,_ and the absorbance was read at 490 nm in a microplate reader (Benchmark, Bio-Rad, Hercules, CA, USA). All assays determined the standard curve, non-specific binding, and maximal binding.

Stanniocalcin protein levels in plasma, CS-homogenates, and incubation medium were quantified by the developed ELISA method for STC.

### 2.4. Ex Vivo Incubation of CS

The corpuscles of Stannius were removed from individual fish (the two CS from each individual were collected) and maintained in 620 μL plasma-like incubation buffer (as described below) gassed with 99.7% O_2_/0.3% CO_2_ for 45 min to allow the tissue to recover after the feasible stress induced by dissection. After CS were transferred individually into the wells of sterile 96-well culture plates (Ref. 83.1837.500, Sarstedt, Newton, NC, USA) containing 75 μL of experimental incubation medium, they were maintained at 25 °C with constant movement in an atmosphere of 99.7/0.3% O_2_/CO_2_. Common incubation medium was described before for this species [35] and contains 173 mM Na^+^, 169 mM Cl^−^, 1 mM Mg^2+^, 3 mM K^+^, 1.5 mM PO_4_^−^, 1 mM SO_4_^2−^, 5 mM HCO_3_^−^, 1.46 mM Ca^2+^, 5 mM glucose and was supplemented with 10 μL mL^−1^ of vitamins (MEM 100× Vitamins, M-6895, Sigma-Aldrich), 20 μL mL^−1^ essential amino acids (MEM 50×, M-5550, Sigma-Aldrich), 10 μL mL^−1^ non-essential amino acids (MEM 100×, M-7145, Sigma-Aldrich), 10 μL mL^−1^ antibiotic (penicillin 10,000 IU mL^−1^; streptomycin 10 mg mL^−1^, P-0781, Sigma-Aldrich), and 20 μL mL^−1^ L-glutamine (200 mM, G-7513, Sigma-Aldrich), at pH 7.80 and osmolality of 360 mOsm kg^−1^.

A time-course survey was performed for the CS ex vivo incubations in the same conditions described before to test the STC secretion rate for 3, 6, and 24 h. Incubations with 0.05% DMSO (D-8779, Sigma-Aldrich) were also performed, as this is the maximum percentage of DMSO employed to pre-dilute drugs in the experiments.

Experimental drug concentrations were obtained from the literature except those specifically described here. The incubation tested calcium effects with 1.46 and 2.92 mM Ca^2+^, corresponding to the free calcium concentrations of control seawater-adapted and hypercalcemic PTHrP-treated seabream [36], respectively. Calcimimetics were used as positive modulators of the CaSR and potential STC production stimulators, like 500 μM Gadolinium (Gd^3+^, G-7532, Sigma-Aldrich) [37], 200 μM Neomycin (N-1876, Sigma-Aldrich) [38], 500 μM Spermine (S-4264, Sigma-Aldrich) [26], and 1 μM R-568 ([(R)-N-(3-methoxy-(ÿ-phenylethyl)-3-(2-chlorophenyl)-1-propylamine hydrochloride], 3815, Tocris Bioscience, Abingdon, UK) [39]. A selective antagonist of the CaSR (a calcilytic drug), the NPS 2143 hydrochloride (2-chloro-6-[(2R)-3–1,1-dimethyl-2-(2-naphthyl)ethylamino-2-hydroxypropoxy]-benzonitrile hydrochloride), was also used at a concentration of 0.1 μM NPS-2143 (3626, Tocris Bioscience, UK) [39]. We tested the putative control of the STC production by second messenger pathways using adenylyl cyclase (AC) inhibitor 100 μM SQ-22536 (S-153, Sigma-Aldrich) and phospholipase C (PLC) inhibitor 10 μM U-73122 (U-6756, Sigma-Aldrich) [40]. The hypercalcemic parathyroid hormone-related protein (PTHrP) was tested at levels of 0 (control), 1, 10, and 100 ng mL^−1^ PTHrP (1–34) [41] covering the physiological range of this hormone in plasma of seabream [36].

All the experimental assays were performed together with a control group, using both CS from a single animal to test experimental vs. control condition ceteris paribus (whenever possible). Repeated independent experiments, with at least 4 to 6 CS per treatment, were also performed to ensure proper statistical robustness of the results.

At the end of the incubation period, CS were transferred to 120 μL 0.6% NaCl and homogenized with a pestle (VWR Pellet Mixer, 431–0100, VWR International), followed by a 10 min, 9000× *g* centrifugation at 4 °C. The supernatant of the CS-homogenates and incubation medium, separately, were immediately frozen at −20 °C until STC analysis.

STC was analyzed in the incubation medium at the end of the ex vivo incubation period to calculate STC secretion (normalized to incubation time and protein quantity in the CS-homogenate). Moreover, total STC was evaluated as the sum of STC quantity in both the incubation medium (and thus secreted from the CS) and that present in the CS at the end of the ex vivo assay. Total STC was employed as a proxy to evaluate STC production in the CS during the incubation time.

### 2.5. CaSR and PTHrP Receptors RT-PCR

Total RNA from the CS was extracted with E.Z.N.A. total RNA isolation kit I (Omega Bio-Tek, Norcross, GA, USA) following the manufacturer’s instructions. The quantity and quality were assessed (Nanodrop 1000 Thermo Scientific, Barrington, IL, USA). Prior to cDNA synthesis, RNA was treated with DNase using a DNA-free kit (Ambion, Life Technologies, Paisley, UK), following the supplier’s instructions. Reverse transcription of RNA into cDNA was carried out using Revert Aid first-strand cDNA synthesis kit (Fermentas, Thermo Scientific, Waltham, MA, USA) following the manufacturer’s instructions, with 500 ng of total RNA in a final reaction volume of 20 µL.

Table 1 shows primer sequences, amplicon sizes, and NCBI accession numbers of the products.

PCR amplifications were performed in a final volume of 15 μL with 7.5 μL SsoFast EvaGreen Supermix (Bio-Rad, UK), ~15 ng cDNA based on RNA input to reverse transcription), and 0.3 µM of each forward and reverse primers. Amplifications were performed in 96-well plates in a T100 thermal cycle (Bio-Rad, UK) with the following protocol: denaturation and enzyme activation step at 95 °C for 2 min, followed by 40 cycles of 95 °C for 5 s, and primer-pair specific annealing temperature for 10 s (Table 1) at 60–55–53 °C for 10 s and 30 s at 72 °C. This protocol runs a product amplification of 131 bp for *casr*, 142 bp *pth1r,* and 250 bp for *pth3r*. 18S ribosomal RNA (*18s*), as the reference gene, generated a 139 bp product. Amplified products (15 μL) were analyzed on 1.5% agarose gel stained with ethidium bromide. Gel images were captured using an AlphaImager Gel Imaging System from Alpha Innotech (San Leandro, CA, USA).

### 2.6. Statistics

Differences between groups were tested by a Student´s *t*-test. One-way ANOVA was employed when different conditions related to the same experiment were tested at the same time, considering those groups as factors of variance. Normality was analyzed using the Kolmogorov–Smirnov´s test. The homogeneity of variances was analyzed by Levene´s test. When ANOVA yielded significant differences, Tukey’s post-hoc test was used to identify significantly different groups. If the results of the same treatment from different independent experiments did not show statistical differences between them, they were pooled and further analyzed as one single group. Statistical significance was accepted at *p* < 0.05. All the results are given as mean ± SEM unless otherwise stated.

## 3. Results

### 3.1. ELISA

We developed an indirect competitive ELISA using purified seabream STC as the standard and a primary rabbit polyclonal antisera raised against it. The coating concentration of CS-homogenates was 2 μg total protein, around 25 ng STC 100 μL^−1^ well^−1^. The primary antiserum against STC was used at a dilution of 1:8000, which gave a maximal OD490 nm of 1.3 and a non-specific background of 0.09–0.13 at OD490 nm well^−1^. The method is reliable and reproducible as the intra- and inter-assay coefficients of variation were 4%, each, calculated from the analysis of seven samples/two times/microplate for the intra-assay coefficient and six samples/four microplates for the inter-assay coefficient. The lower detection limit of the assay was 2.6 ng STC well^−1^ (52 ng mL^−1^). Dilution curves constructed with plasma samples and CS-homogenates showed parallel displacements compared to the standard dilution curve (Appendix A).

### 3.2. STC Secretion and CS Content

The STC secretion rate, when normalized to time and amount of protein, does not present variations in ex vivo incubations of CS for 3, 6, or 24 h (Appendix A) (one-way ANOVA followed by a Tukey´s test, *p* < 0.05). Since proper dilution of some of the chemicals employed in this study required the addition of DMSO, we also performed ex vivo incubations of CS with a final concentration of 0.05% DMSO (*v*/*v*) for 3, 6, and 24 h. The results show that the STC secretion rate decreased during incubation time in vitro, with significant differences between the groups at 3 and 24 h (one-way ANOVA followed by a Tukey´s test, *p* < 0.05). When comparing control versus DMSO treated groups (Student´s *t*-test, *p* < 0.05), no differences were observed in any of the groups tested. In this line, the secreted quantity of STC after 3 h of incubation edges the detection limits of the ELISA method in some samples. We, therefore, choose an incubation time of 6 h for the experiments. Thus, the average STC secretion rate from the CS of control gilthead seabream (84.2 ± 1.9 g weight individuals) is 0.18 ± 0.03 ng STC μg prot^−1^ h^−1^ (*n* = 52) (mean ± SEM). The average content of STC in the CS at the moment of sampling was 11.17 ± 1.31 ng STC μg prot^−1^ (*n* = 4), which is in good agreement with the STC content of the CS of the control groups after six hours of incubation (10.33 ± 0.85 ng STC μg prot^−1^, *n* = 52) (Student´s *t*-test, *p* = 0.77). Plasma STC concentration in seabream was 1693 ± 81 ng STC mL^−1^ (*n* = 42 fish). Assuming that the plasma of gilthead seabream is 2.00% of its total body weight (blood occupies 2.9% of the total body weight [43], while the normal hematocrit for seabream is 31% [44]), we have calculated the total amount of STC within the CS (in both together, from a single fish) of each animal compared to the total circulating STC in the plasma, which results in 9.8 ± 1.0% (*n* = 42).

### 3.3. CaSR and PTHrP Receptors

Figure 1 shows the amplification of specific RT-PCR and confirms the presence of the *casr* and the PTHrP receptors *pth1r* and *pth3r* in the CS of the gilthead seabream.

### 3.4. Effect of CaSR Agonist and Antagonist Molecules

Figure 2A shows the percentage (%) of STC secreted by seabream´s CS incubated for six hours with normal (1.46 mM Ca^2+^) and high calcium (2.92 mM Ca^2+^) levels. A significant increase in STC release (Student´s *t*-test, *p* < 0.05) was observed when incubating in the presence of 2.92 mM Ca^2+^ (657 ± 83% compared to the control-normal calcium group). Figure 2B shows the % of total STC quantity (ng STC μg prot^−1^) present in the CS and the incubation medium, together, for the experimental group respecting the control group. No differences were found between both groups (Student´s *t*-test, *p* < 0.05).

The effect of a CaSR antagonist is shown in Figure 3. Incubation with a high concentration of calcium (2.92 mM Ca^2+^) and the addition of 0.1 μM of the CaSR antagonist NPS-2143 (Figure 3A) decreased STC release from the CS to 50 ± 10% (mean ± SEM) of the control group (with 2.92 mM Ca^2+^ alone), with significant differences (Student´s *t*-test, *p* < 0.05). Moreover, the calcilytic NPS-2143 decreased significantly (Student´s *t*-test, *p* < 0.05) the total amount of STC compared to the control-high [Ca^2+^] group (49 ± 4%) (Figure 3B). 

The effect of different CaSR agonists (R-568, neomycin, gadolinium, and spermine) was also tested (Figure 4). Significant increases when compared to the control CS STC release (Student´s *t*-test, *p* < 0.05) are shown in Figure 4A when incubated with calcimimetics R-568 (260 ± 56%), neomycin (381 ± 104%) and gadolinium (166 ± 6%). Spermine, at a concentration of 500 μM, did not affect the STC secretion rate (Student´s *t*-test, *p* > 0.05). The calcimimetics, neomycin and gadolinium, also increased significantly the total amount of STC (125 ± 7% and 200 ± 37%, respectively) of the incubated CS when compared to the control group (Figure 4B, Student´s *t*-test, *p* < 0.05).

### 3.5. PTHrP Downregulates STC Secretion

Figure 5A shows the inhibitory effect of increasing PTHrP concentrations on the STC secretion at normal (1.46 mM) calcium levels. PTHrP inhibits STC release from the CS in a dose-dependent manner, with significant differences between those groups submitted to 0 ng PTHrP mL^−1^ and 100 ng PTHrP mL^−1^ (one-way ANOVA followed by a Tukey´s test, *p* < 0.05). In this sense, the calculated effective dose 50 (ED50) was 17.51 ng PTHrP mL^−1^. The % of total STC (Figure 5B) showed variations in any of the groups tested (one-way ANOVA followed by a Tukey´s test, *p* > 0.05).

Figure 6 reveals that the tmAC inhibitor SQ-22536 and the PLC inhibitor U-73122, separately, slightly increased STC secretion (Figure 6A) when it is inhibited with 100 ng PTHrP mL^−1^, without being statistically different from the untreated-control or the PTHrP-treated groups (one-way ANOVA, followed by a Tukey´s test, *p* < 0.05). Moreover, using both inhibitors together abolished the inhibitory effect of PTHrP (107 ± 28% respecting the untreated-control group). No differences in STC production (Figure 6B) were observed between treatments (one-way ANOVA, followed by a Tukey´s test, *p* > 0.05).

### 3.6. Combined Effects of High Calcium and PTHrP

Figure 7 shows the combined effects of high (2.92 mM) calcium levels and increasing PTHrP concentrations. STC secretion (Figure 7A) or production (Figure 7B) are not significantly affected by any of the PTHrP concentrations tested (one-way ANOVA followed by a Tukey´s test, *p* > 0.05).

## 4. Discussion

In the present study, we demonstrate the regulatory action of calcium and PTHrP on the release of stanniocalcin from the corpuscles of Stannius, a specific bony fish gland and endocrine tissue. These effects involve the participation of a calcium-sensing receptor and imply an unexpected role of the PTHrP receptors, which, based on transcriptomic evidence, are supposedly absent from the CS of a teleost fish [45]. In this sense, the antagonistic hormone of the STC, the PTHrP, exerts a direct down-regulation of the STC secretion through the participation of at least two PTHrP receptors, likely receptor1 and receptor3.

A CaSR is present in the CS glands of the gilthead seabream, an observation that reinforces previous studies in trout [18], flounder [17], and zebrafish [19]. The first two studies described a fast 9-fold and 3-fold increase in the plasma STC levels in response to intraperitoneal injections of the calcimimetics NPS-467 or R-568. The resemblance between these in vivo and our ex vivo results is evident as we show a 6.5-fold increase in ex vivo STC secretion in response to high calcium concentrations for six hours. This result is reinforced by the stimulation ranges of STC secretion of a 1.6 to 3.8-fold increase in response to the tested calcimimetics used in our study. Previous studies with rainbow trout and European eel also show increased STC release to the plasma in response to CaCl_2_ administration [46]. As those results came from in vivo experiments, side effects or interfering alternative routes in the regulation of STC secretion need consideration. However, our ex vivo approach negates those circumstances and facilitates a direct understanding of the effects at the level of the gland. Thus, the stimulation of STC secretion not only by Ca^2+^ but also through the calcimimetics R-568, neomycin, and gadolinium indicates the direct involvement of CaSR in STC secretion regulation.

In addition, calcimimetics, neomycin and gadolinium, also increase the total percentage of available STC, pointing to a possible enhancement of both, increase in secretion and stimulation of hormone synthesis. Furthermore, the allosteric antagonist of the CaSR NPS-2143 down-regulates STC secretion and the STC synthesis in the CS gland. In this sense, using a calcilytic compound in flounder also decreased plasma STC levels to half of the control after treatment with EGTA [17], supporting our results. Thus, the involvement of CaSR in the secretion and synthesis of STC is with this demonstrated in the seabream.

Previous studies show that STC and PTHrP exert functional antagonistic actions at different levels [3,22], but to date, and as far as we are aware, there is no experimental evidence for a putative reciprocal control of their production/secretion. The CS gland, in this sense, offers a unique model to study interactions between both hormones through the analysis of STC synthesis and release. Through next-generation sequencing techniques, there is a reported lack of PTHrP receptor transcripts in the CS of the Japanese eel [45]. However, in the present study, we have demonstrated with a specific RT-PCR approach the presence of two PTHrP receptors (*pth1r* and *pth3r*) in the CS of the seabream. Moreover, ex vivo incubations of the CS with PTHrP revealed its inhibitory action on STC release in a dose-dependent fashion. There are no previous reports of this interaction in fish or mammals. This knowledge gap also extends to the mammalian counterparts of the PTHrP and STC, the PTH, and the calcitonin hormone, respectively. Hence, these results highlight the importance of calcium homeostasis regulation, as its intertwined pathways are mutually regulated. Therefore, we calculated the PTHrP ED50 for STC secretion, 17.51 ng mL^−1^. Normal levels of PTHrP in the plasma of seabream range from 2.50 ± 0.29 ng mL^−1^ [47] to 0.86 ± 0.25 ng mL^−1^ [48]; therefore, a slight but constant inhibition of STC release from the corpuscles of Stannius in seabream takes place at circulating levels of PTHrP.

The PTHrP effects described in this study are likely achieved through specific receptors and second messenger routes. The inhibitory effects of PTHrP in STC secretion are mediated by the activation of a phospholipase C (PLC) and transmembrane adenylyl cyclase (tmAC), as specific inhibitors of both enzymes rescue the inhibition caused by the PTHrP towards the STC secretion at the same time (but not separately). In this sense, gilthead seabream PTH3R (sbPTH3R) seems only to activate the cAMP pathway [32], thus suggesting that the PTH1R in the CS primarily activates the PLC signaling cascade. Our results are in good agreement with previous studies in other species ranging from mammals to fish, as the PTH/PTHrP receptors are closely linked to PKA and PLC signaling pathways [2,28,31] unless their affinities to one or other second messenger routes depend on the specific type of PTHrP receptor. However, both routes are related to the inhibition of the STC secretion in the CS of the gilthead seabream through the action of the PTHrP. Moreover, in zebrafish, a gene duplication produced two distinct PTH molecules with different receptor affinities [49]. If seabream also presents different effectors of the two PTHRs located in the CS, the overall receptor selectivity may present different action patterns depending on substrate binding.

Interestingly, the CS presents two types of granulated cells [50,51], which opens a possibility for different regulatory mechanisms. To understand if both PTHrP receptors are present in the same pool of STC secretory cells or if their inhibitory actions are related to differentiated pools requires further investigation. However, if differentiated groups of STC secretory cells exist, it may indicate that there are other regulatory agents in addition to calcium. In this sense, earlier work on STC postulated that the nervous system might act as a promotor of STC secretion [46]; however, we should not rule out other unknown external factors. The inhibitory action of the sbPTH3R through the production of cAMP matches with the described inhibition of cAMP accumulation due to CaSR stimulation [25,27]. Thus, the PTH3R effects in this species could be related to inhibiting the STC-secretion stimulated by basal Ca^2+^ levels through a tmAC. Moreover, our results also show that at high plasma Ca^2+^ concentrations, the PTHrP receptors do not inhibit the STC secretion. The reason for that is still not understood, but it could imply the presence of another mechanism to regulate secretion in the CS.

## 5. Conclusions

In conclusion, using ex vivo approaches and RT-PCR techniques, we have demonstrated consistent STC release from the CS of gilthead seabream controlled by calcium levels and its antagonist hormone, the PTHrP. Three types of membrane receptors are thus mediating this action: CaSR, PTH1R, and PTH3R. The inhibition exerted by both PTHrP receptors could imply alternative pathways, maybe due to differentiated pools of STC secretory cells. Further studies to elucidate whether or not this hypothesis is true are necessary. The present study demonstrates that the regulation of antagonist hormones such as STC and PTHrP in teleost fish is more closely related than expected.

## Figures and Tables

**Figure 1 biology-11-00863-f001:**
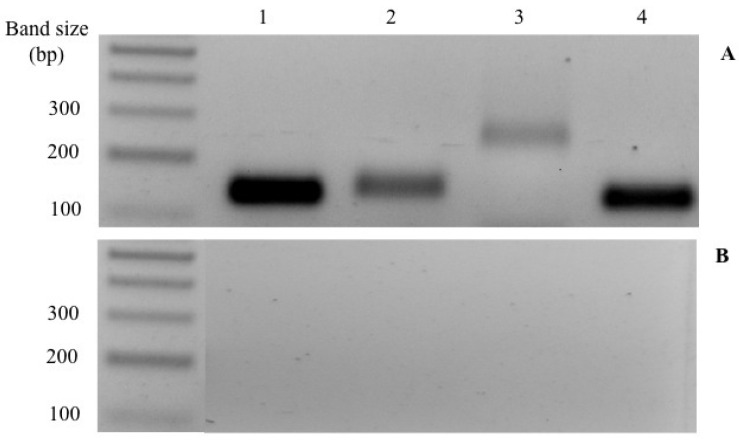
Specific RT-PCR was used to amplify seabream *pth1r* (2), *pth3r* (3), and *casr* (4) in mRNA extracts from the corpuscles of Stannius (**A**). *18s* (1) was used as a reference to control the quantity and quality of cDNA included in PCR reactions. The size of the bands is 139 bp for *18s*, 142 bp for *pth1r*, 250 bp for *pth3r,* and 131 bp for *casr*. A negative control of every reaction is displayed in (**B**).

**Figure 2 biology-11-00863-f002:**
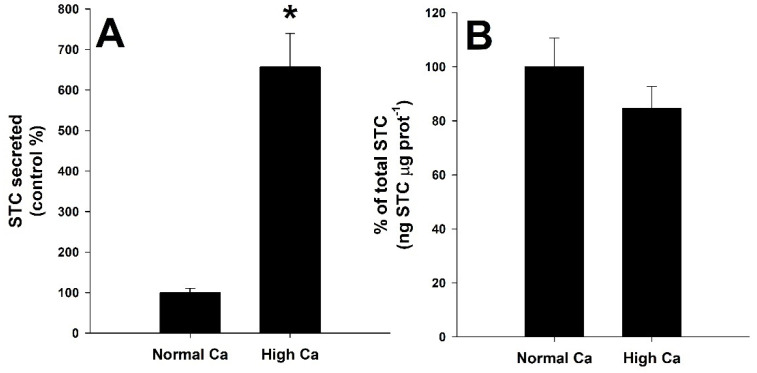
Stanniocalcin (STC) ex vivo secretion (**A**) and production (**B**) from the corpuscles of Stannius of the gilthead seabream, *Sparus aurata*, after treatment with normal calcium concentration (1.46 mM Ca^2+^) or high calcium (2.92 mM Ca^2+^). Basal STC secretion for the control group is 0.16 ± 0.04 ng STC μg prot^-1^ h^−1^. Values are referred to as % from the control group (mean ± SEM) for single independent experiments (*n* = 19 per group). Asterisks (*) indicate significant differences from the control group (Student´s *t*-test, *p* < 0.05).

**Figure 3 biology-11-00863-f003:**
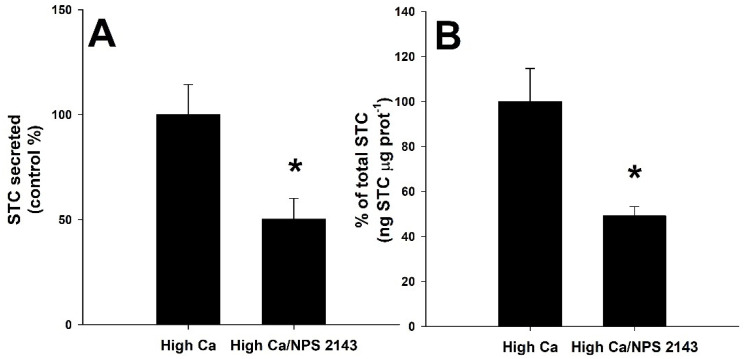
Stanniocalcin (STC) ex vivo secretion (**A**) and production (**B**) from the corpuscles of Stannius of the gilthead seabream, *Sparus aurata*, incubated with high calcium (2.92 mM Ca^2+^) in the absence or presence of the CaSR antagonist 0.1 μM NPS-2143. Values are presented as % of the control-high calcium group (1.12 ± 0.16 ng STC μg prot^−1^ h^−1^, mean ± SEM) for independent experiments (*n* = 6 per group). Asterisks (*) indicate significant differences from the control group (Student´s *t*-test, *p* < 0.05).

**Figure 4 biology-11-00863-f004:**
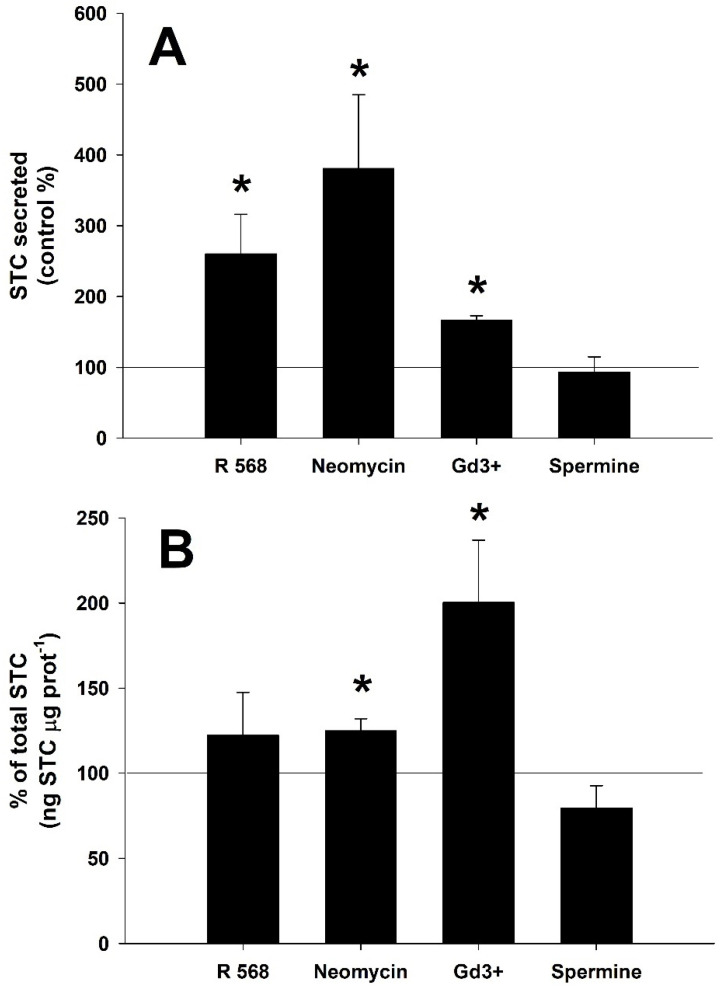
Stanniocalcin (STC) ex vivo secretion (**A**) and production (**B**) from the corpuscles of Stannius of the gilthead seabream, *Sparus aurata*, after treatment with different calcium-sensing receptor (CaSR) putative agonists: 1 μM R-568, 200 μM neomycin, 500 μM gadolinium (Gd^3+^) and 500 μM spermine. Basal STC secretion for the control group is 0.16 ± 0.03 ng STC μg prot^−1^ h^−1^ (*n* = 24). Values are presented as % of the control group (horizontal line) for single independent experiments (mean ± SEM, *n* = 6 per group). Asterisks (*) indicate significant differences from the control group (Student´s *t*-test, *p* < 0.05).

**Figure 5 biology-11-00863-f005:**
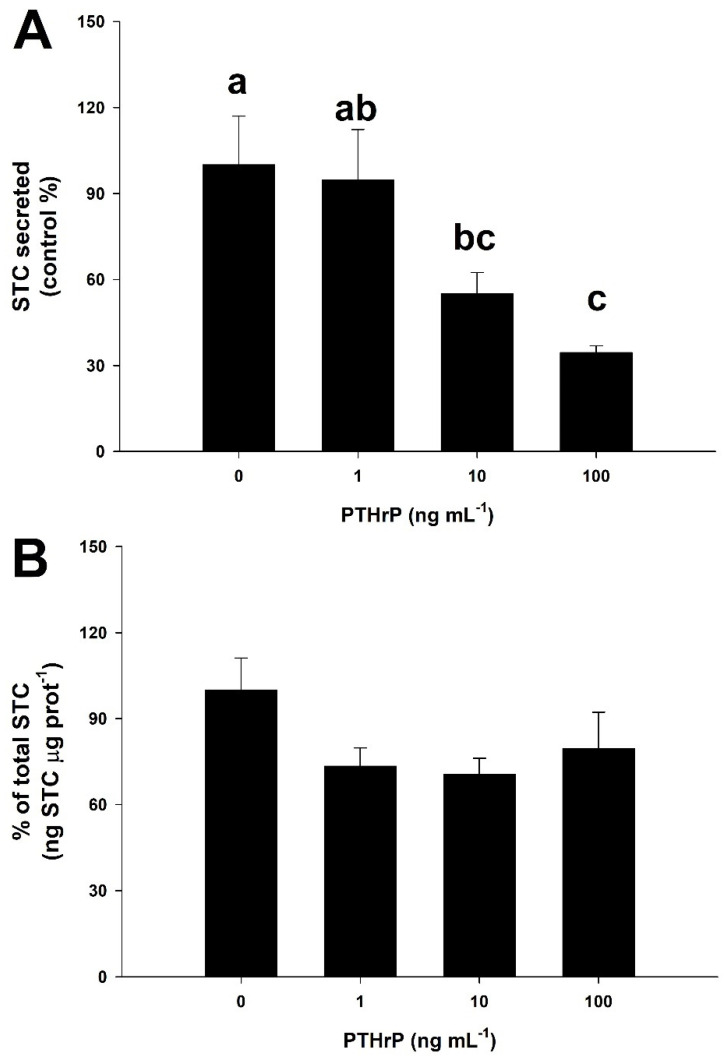
Dose-response effects of PTHrP on STC ex vivo secretion (**A**) and production (**B**) from the corpuscles of Stannius of the gilthead seabream at normal (1.46 mM Ca^2+^) calcium concentration. Basal STC secretion for the 0 ng PTHrP mL^−1^ group is 0.37 ± 0.06 ng STC μg prot^−1^ h^−1^ (*n* = 6). Values are presented as % of the control group for single independent experiments (mean ± SEM, *n* = 6 per group). Different letters indicate significant differences between groups (one-way ANOVA followed by a Tukey´s post-hoc test, *p* < 0.05).

**Figure 6 biology-11-00863-f006:**
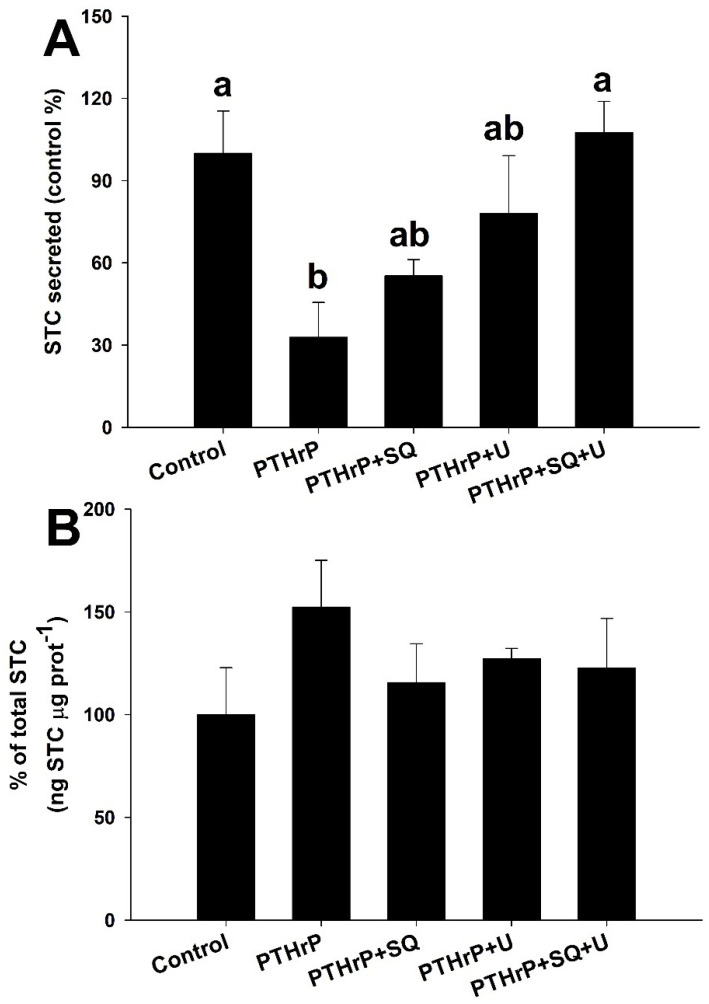
STC ex vivo secretion (**A**) and production (**B**) from the corpuscles of Stannius of the gilthead seabream treated with 100 ng mL^−1^ PTHrP alone or with combinations of the adenylyl cyclase inhibitor SQ-22536 (100 μM, SQ) and the phospholipase C inhibitor U-73122 (10 μM, U). Control group presents 1.46 mM Ca^2+^ and no PTHrP or specific inhibitors (basal STC secretion of 0.19 ± 0.03 ng STC μg prot^−1^ h^−1^, mean ± SEM). Different letters indicate significantly different groups (*n* = 6 per group, one-way ANOVA followed by a Tukey´s post-hoc test, *p* < 0.05).

**Figure 7 biology-11-00863-f007:**
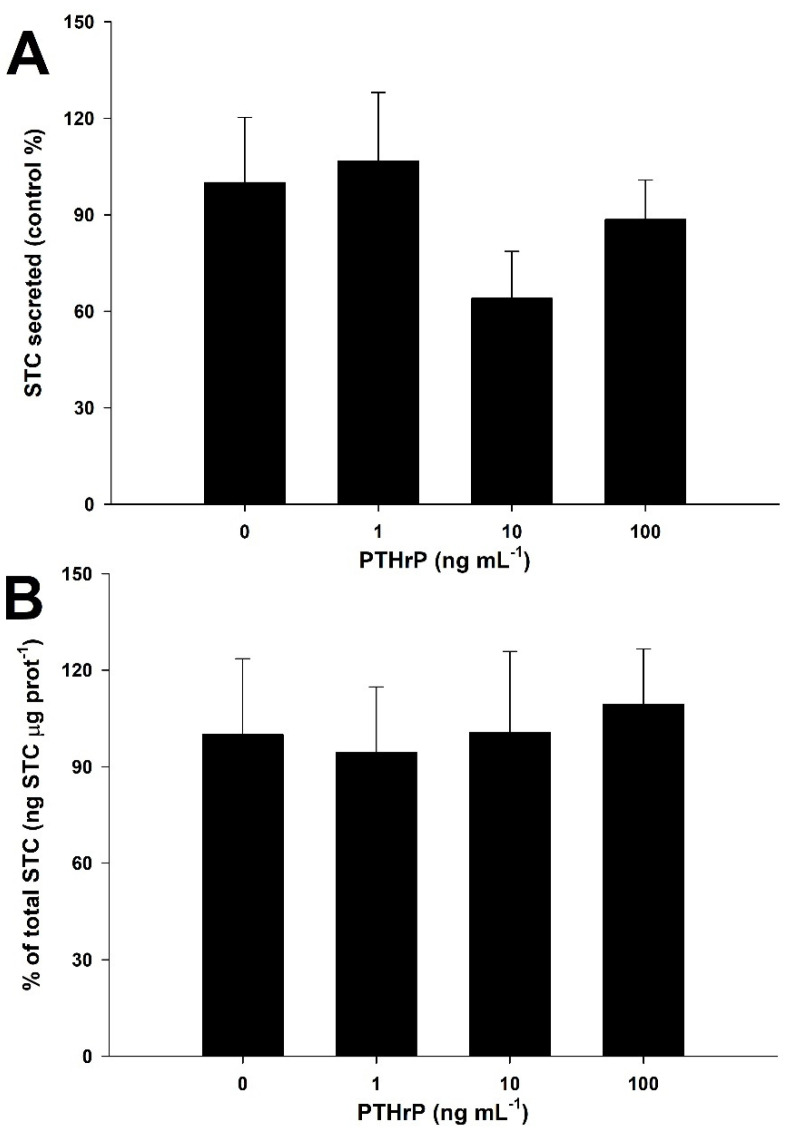
Dose-response effects of PTHrP on STC ex vivo secretion (**A**) and production (**B**) from the corpuscles of Stannius of the gilthead seabream at high (2.92 mM Ca^2+^) calcium concentration. Basal STC secretion for the 0 ng PTHrP mL^−1^ group is 0.82 ± 0.17 ng STC μg prot^−1^ h^−1^ (mean ± SEM). No significant differences between groups were observed (*n* = 9 per group, one-way ANOVA followed by a Tukey´s post-hoc test, *p* > 0.05).

**Table 1 biology-11-00863-t001:** Details of primers used for RT-PCR.

Gene	Sequence (5′–3′)	Ta (°C)	Amplicon (bp)	Accession nº
*Casr* ^a^	Fw AGAGTTCTTACAGCACGTCCAAC	60	131	AJ289717
	Rv CTAGTGCTGCCATCTCACTTTC			
*pth1r* ^b^	Fw TCACCAACGTCACTGCCAGAGGA	55	142	AJ619024
	Rv GTCCCGACGAGGGTATCGAGTT			
*pth3r* ^b^	Fw ACATCCACATTCACTTCTTCAC	53	250	AY547261
	Rv GATGAGGGCCACAGGTAGT-			
*18s*	Fw AACCAGACAAATCGCTCCAC	60	139	AY993930
	Rv CCTGCGGCTTAATTTGACTC			

Forward (Fw) and reverse (Rv) primers; annealing temperature (Ta); base pairs (bp). ^a^ [24]; ^b^ [42].

## Data Availability

The data presented in this study are available on request from the corresponding authors.

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
