# Peer review of "Regulation of Stanniocalcin Secretion by Calcium and PTHrP in Gilthead Seabream (Sparus aurata)"

_biology, 2022, doi:10.3390/biology11060863_

Round 1

Reviewer 1 Report

The study of Ruiz-Jarabo et al, represents that the regulation of antagonist hormones such as STC and PTHrP in teleost fish is more closely related than expected. The paper is well written in a scientific circumstance just minor corrections to be accepted for publication.

Q1 Could you include the materials and methods a photo showing the right place of corpuscles of stannius in seabream

Q2 Data support sections 3.1 and 3.2 could be supplementary materials better than data not shown.

Q3 Corpuscles of stannius in fish could refer to as a gland or as endocrine tissue (line 348).

Q4 Could you use lowercase letters to indicate significance in Fig 5 and 6, to prevent interruption that could occur due to the use of uppercase letters (A, and B) to mark of the figures.

Author Response

REVIEWER 1

Comments and Suggestions for Authors

The study of Ruiz-Jarabo et al, represents that the regulation of antagonist hormones such as STC and PTHrP in teleost fish is more closely related than expected. The paper is well written in a scientific circumstance just minor corrections to be accepted for publication.

Q1 Could you include the materials and methods a photo showing the right place of corpuscles of stannius in seabream.

We have included a photo (Figure S1), as suggested.

Q2 Data support sections 3.1 and 3.2 could be supplementary materials better than data not shown.

This information has been included as Figures S2 and S3, as suggested.

Q3 Corpuscles of stannius in fish could refer to as a gland or as endocrine tissue (line 348).

The suggestion has been included.

Q4 Could you use lowercase letters to indicate significance in Fig 5 and 6, to prevent interruption that could occur due to the use of uppercase letters (A, and B) to mark of the figures.

Done, as suggested.

Reviewer 2 Report

This is a well-written manuscript and perfectly sound. The results are significant in improving our understanding of calcium homeostasis in teleosts.

The main issue that the reviewer has is in the description of the measurements of the % of total STC vs the amount of STC secreted. The STC secreted is measured by isolating the corpuscles of Stannius and incubating them in various medium in order to measure how much STC is secreted (into the medium?) after 6h of incubation by the CS while the % of total STC is measurement of how much STC is contained in the glands(produced but not secreted yet) and medium (secreted) at the same time after 6h of incubation? How are those measurements done? Where the CS removed before the measurements of the STC secreted?

Minors revisions are listed below with the line number. Those are mainly spelling issues and formatting issues:

- Throughout the manuscript, the way words are splits across 2 lines is sometimes incorrect, see L19, L142, L146, L149, L151, L233, L265, L352.

- L91: "in" should be removed to read: "transported to the Campus facility"

-L92: This sentence is a little long and could benefit from being split into 2-3 sentences. It could read: ".... with a biological filter. The water temperature was maintained at 23-25 °C. The fish were exposed to a natural photoperiod, grouped in tanks at a density of 4.5 kg m-3...."

- The indication of the dilutions of chemicals as an indication of how much was used is unprecised, especially considering that manufacturers may change the initial concentration of the chemicals they are selling. The reviewer suggest to either indicated only the actually concentration used or to at least indicate the initial concentration of the chemicals. This should be done lines 114, 125, 128 and 222.

- L129: "followed by an incubating similar as..." It should be "incubation".

- L142-145: the sentence should read "...experimental incubation medium, they were maintained at..."

- L152: authors should add "at" before pH.

-L198: "transcription" is mispelled.

-L225: missing spaces around / between times and microplate, for consistency.

- References list:

*there is an issue with the numbering appearing twice

*the charges of ions are not always displayed as exponent Ca2+, instead of Ca2+ for example. L454, 499, 501, 507.

*the name of the journals are not always abbreviated: L486, 504, 536, 571.

Author Response

REVIEWER 2

Comments and Suggestions for Authors

This is a well-written manuscript and perfectly sound. The results are significant in improving our understanding of calcium homeostasis in teleosts.

The main issue that the reviewer has is in the description of the measurements of the % of total STC vs the amount of STC secreted. The STC secreted is measured by isolating the corpuscles of Stannius and incubating them in various medium in order to measure how much STC is secreted (into the medium?) after 6h of incubation by the CS while the % of total STC is measurement of how much STC is contained in the glands(produced but not secreted yet) and medium (secreted) at the same time after 6h of incubation? How are those measurements done? Where the CS removed before the measurements of the STC secreted?

We have included a paragraph at the end of section 2.4 to better explain this part. It now reads as:

STC was analyzed in the incubation medium at the end of the ex vivo incubation period to calculate STC secretion (normalized to incubation time and protein quantity in the CS-homogenate). Moreover, total STC was evaluated as the sum of STC quantity in both the incubation medium (and thus secreted from the CS) and that present in the CS at the end of the ex vivo assay. Total STC was employed as a proxy to evaluate STC production in the CS during the incubation time.

Minors revisions are listed below with the line number. Those are mainly spelling issues and formatting issues:

- Throughout the manuscript, the way words are splits across 2 lines is sometimes incorrect, see L19, L142, L146, L149, L151, L233, L265, L352.

The splits are automatically done by the “Biology template” and the authors can not modify them. We will translate this concern to the Editor in charge.

- L91: "in" should be removed to read: "transported to the Campus facility"

Done as suggested.

-L92: This sentence is a little long and could benefit from being split into 2-3 sentences. It could read: ".... with a biological filter. The water temperature was maintained at 23-25 °C. The fish were exposed to a natural photoperiod, grouped in tanks at a density of 4.5 kg m-3...."

We would like to thank the Reviewer for this suggestion. We included all suggested changes.

- The indication of the dilutions of chemicals as an indication of how much was used is unprecised, especially considering that manufacturers may change the initial concentration of the chemicals they are selling. The reviewer suggest to either indicated only the actually concentration used or to at least indicate the initial concentration of the chemicals. This should be done lines 114, 125, 128 and 222.

We appreciate this comment, and the Reviewer is right. However, the primary antisera (anti-STC developed in rabbit) is not commercially available and was developed by our research group (Gregório et al., 2014) so that we have no further information about the initial concentration of specific anti-STC antibodies. Regarding the secondary antisera (A9169, Sigma-Aldrich): all available information can be found in the official webpage of Sigma-Aldrich (https://www.sigmaaldrich.com/ES/es/product/sigma/a9169).

- L129: "followed by an incubating similar as..." It should be "incubation".

Done, as suggested.

- L142-145: the sentence should read "...experimental incubation medium, they were maintained at..."

Done as suggested.

- L152: authors should add "at" before pH.

Done as suggested.

-L198: "transcription" is mispelled.

Thanks for the sharp revision. We have modified the word as suggested.

-L225: missing spaces around / between times and microplate, for consistency.

Done as suggested. Thanks for the comment.

- References list:

*there is an issue with the numbering appearing twice

Corrected.

*the charges of ions are not always displayed as exponent Ca2+, instead of Ca2+ for example. L454, 499, 501, 507.

Corrected, as suggested.

*the name of the journals are not always abbreviated: L486, 504, 536, 571.

Thanks for the comment. We have modified all journal names accordingly.